# Oral Administration of *Bacillus subtilis* Subunit Vaccine Significantly Enhances the Immune Protection of Grass Carp against GCRV-II Infection

**DOI:** 10.3390/v14010030

**Published:** 2021-12-24

**Authors:** Yang Gao, Xingchen Huo, Zhensheng Wang, Gailing Yuan, Xiaoling Liu, Taoshan Ai, Jianguo Su

**Affiliations:** 1Department of Aquatic Animal Medicine, College of Fisheries, Huazhong Agricultural University, Wuhan 430070, China; gaoyang801314@163.com (Y.G.); huoxingchen2020@webmail.hzau.edu.cn (X.H.); wangzhensheng19@163.com (Z.W.); yuangailing@mail.hzau.edu.cn (G.Y.); liuxl@mail.hzau.edu.cn (X.L.); 2Laboratory for Marine Biology and Biotechnology, Pilot National Laboratory for Marine Science and Technology, Qingdao 266237, China; 3Hubei Hongshan Laboratory, Huazhong Agricultural University, Wuhan 430070, China; 4Engineering Research Center of Green Development for Conventional Aquatic Biological Industry in the Yangtze River Economic Belt, Ministry of Education, Wuhan 430070, China; 5Wuhan Chopper Fishery Bio-Tech Co., Ltd., Wuhan Academy of Agricultural Science, Wuhan 430207, China; scs@wuhanagri.com

**Keywords:** grass carp (*Ctenopharyngodon idella*), grass carp reovirus, *Bacillus subtilis*, oral vaccine, VP56 protein

## Abstract

Grass carp reovirus (GCRV) is a severe virus that causes great losses to grass carp culture every year, and GCRV-II is the current popular and fatal strain. VP56, fibrin on the outer surface of GCRV-II, mediates cell attachment. In this study, we firstly divided the VP56 gene into four fragments to screen the optimal antigen by enzyme-linked immunosorbent assay and neutralizing antibody methods. The second fragment VP56-2 demonstrates the optimal efficiency and was employed as an antigen in the following experiments. *Bacillus subtilis* were used as a carrier, and VP56-2 was expressed on the surface of the spores. Then, we performed the oral immunization for grass carp and the challenge with GCRV-II. The survival rate was remarkably raised, and mRNA expressions of *IgM* were significantly up-regulated in spleen and head kidney tissues in the *B. s*-CotC-VP56-2 group. Three crucial immune indexes (complement C3, lysozyme and total superoxide dismutase) in the sera were also significantly enhanced. mRNA expressions of four important genes (*TNF-α*, *IL-1β*, *IFN1* and *MHC-II*) were significantly strengthened. Tissue lesions were obviously attenuated by histopathological slide examination in trunk kidney and spleen tissues. Tissue viral burdens were significantly reduced post-viral challenge. These results indicated that the oral recombinant *B. subtilis* VP56-2 subunit vaccine is effective for controlling GCRV infection and provides a feasible strategy for the control of fish virus diseases.

## 1. Introduction

In China, grass carp (*Ctenopharyngodon idella*) is an important economic fish, and grass carp hemorrhagic disease is an important disease that endangers grass carp farming industry, causing the grass carp farming to suffer a great loss every year. Grass carp hemorrhagic disease is caused by grass carp reovirus (GCRV) [1]. At present, based on sequence comparison and analysis, known GCRV strains can be divided into three different types. They are GCRV-I (GCRV 873, GCRV GZ1208, etc.), GCRV-II (GCRV-HZ08, GCRV-GD108, GCReV-109, etc.) and GCRV-III (HGDRV) [2,3,4,5]. The most common and most harmful strain is GCRV-II [6]. GCRV is a double-stranded RNA (dsRNA) virus. The genome consists of 11 segments (called s1–s11), which are wrapped in a multi-layered icosahedral capsid [2,7]. It encodes seven structural proteins (VP1–VP7) and five non-structural proteins [8]. However, most research has focused on GCRV-I, and there is limited information about GCRV-II [2].

In the development of GCRV vaccines, most are developed around the GCRV-I outer capsid protein, such as subunit and DNA vaccines made by VP5 and VP7 [6,9,10,11]. However, epidemiological investigations have shown that GCRV-II is the main virus type in the current period, but the vaccines based on the GCRV-II are scarce [3,12]. VP56 is a protein encoded by the S7 segment of the GCRV-II. This fragment has 27% of the adenovirus fibrin sequence characteristics in the amino acid sequence 100–300 region, and VP56 has a predicted helical region at the N-terminus. Next is the β-helix motif, similar to other known fibrin proteins [5,13]. Some scholars predict that VP56 may be used as a viral adhesion protein to bind to the host cell membrane [14,15]. Studies have also shown that VP56 induces K48 chain ubiquitination and IRF7 degradation to inhibit the production of interferon (IFN), so VP56 is predicted to be fibrin that may be immunogenic on the surface of the outer capsid [16].

Immune prevention is one of the most effective methods for viral diseases. Many types of vaccines for GCRV have been developed, such as inactivated vaccines, attenuated vaccines, subunit vaccines and DNA vaccines [13,17]. However, these vaccines might be not safe enough and unstable and can easily lead to the recovery of the disease from the mutation of the virus [18]. DNA vaccines have the advantages of ease of production, genetic stability without cold chain requirements and the activation of humoral and cellular immunity. However, the cost of manual injection is high; it is not easy to use in actual production; and the long-term expression of plasmids in the body is unclear. Subunit vaccines are usually prepared from viral capsid proteins and bacterial glycoproteins, which have enhanced immune protection [19,20]. Compared with other vaccines, subunit vaccines can provide effective antigens and stimulate the immune response without introducing pathogens. Due to the low cost and safety, it can be produced as a multivalent vaccine [17].

Since the first report on surface display systems in 1985 [21], some surface display systems, such as bacteriophages [22], yeasts [23] and bacteria [24], have been reported. However, these systems have not been found to be suitable for harsh environmental conditions. On the other hand, bacterial spores have the ability to maintain sporulation and germination cycles in extreme environments, making them ideal candidates for surface display technologies [25]. Among the bacteria that can form spores, *Bacillus subtilis* can spore and survive under extreme conditions, such as mechanical extrusion, high temperature, acid, alkali and other extreme environments. *B. subtilis* is a Gram-positive-bacteria. Its probiotic characteristics and its recognized safety (GRAS) certification are considered attractive for surface display technology [26]. It has recently been used as an effective oral carrier for bacterial and parasitic vaccines as well as food supplements and probiotics for human and animal gastrointestinal diseases. Since *B. subtilis* spore surface display (BSSD) technology was first applied to combine the tetanus toxin gene with the gene encoding the spore ankyrin CotB (TTFC) [25,27], it has been used in many oral vaccines. In previous studies, some scholars expressed the Sip protein on *B. subtilis* [28], which can protect Streptococcus agalactiae from infection to a certain extent. It is also often used in the development of aquatic oral vaccines [29,30,31].

*IgM* is the most abundant immunoglobulin in blood and mucus-associated lymphoid tissues (MALT). It is also considered to be the most important immunoglobulin in the immune system of osteogenic fish [32,33,34]. *IgM* is the main immunoglobulin in osteogenic fish and widely exists in all parts of the body to participate in the collective immune response. It is an indispensable immunoglobulin in the body, and after being antiviral vaccine immunization, public B cells are activated in the body to produce related immune substances in the body’s immune response, *IgM* is one of them. In our study, the method is oral immunization vaccination, which can act directly on the grass carp in the gut, causes minimal harm to the body, and is easy to operate. It is also the main aspect of aquatic vaccine research in the future.

Oral antigens lead to stimulation of innate and adaptive immune responses, in which intestinal absorption is expected [35]. Similar to systemic immune response, antigen-presenting cells (APCs), such as monocytes and macrophages, and intraepithelial B and T lymphocytes of fish gut propria play an important role in antigen uptake, which is related to the expression of different cytokines [36]. The cell-mediated immune response in fish mucosa is expressed by CD4 and CD8 genes in the mucosal immune response [37]. In addition, mucosal immunity produces antibodies that protect the intestinal mucosa from microbial invasion [37,38]. However, the challenge remains to demonstrate the mechanisms that mediate these processes in mucosal immunity.

In this study, we analyzed the potential epitopes of VP56 and used BSSD to evaluate the protection against GCRV-II infection. Firstly, we analyzed the immunogenicity of VP56 and divided it into 4 segments, expressed in *Escherichia coli*, respectively, and their potential epitopes were analyzed by ELISA and neutralizing antibody experiments. Secondly, *B. subtilis* coat protein C was used as the fusion partner to connect VP56-2 to *B. subtilis* WB600. The optimal expression time of the recombinant strain was screened, and then immunofluorescence experiment and flow cytometry were used to determine its expression on the spore surface. Finally, a challenging experiment in vivo was carried out to evaluate its protective effect against GCRV-II infection.

## 2. Materials and Methods

### 2.1. Fish, Bacteria and Virus

Healthy grass carp, weighing about 15 g, were obtained from a fish farmer in Huang Gang, Hubei Province, China. The animals acclimatized for one week before experiments. GCRV was examined from random individuals by qRT-PCR to eliminate the possibility that these fish were virus carriers.

Plasmid constructs for subcloning experiments were carried out with *E. coli* strain DH5α or BL21(DE3) (Invitrogen, Carlsbad, CA, USA). *B. subtilis* WB600 were bought from Miao Ling Plasmid Sharing Platform (Wuhan, China). The GCRV-II used in this experiment is GCRV 097 strain, stored in our laboratory.

Animal studies were conducted in accordance with the ethical guidelines and protocols approved by the Huazhong Agricultural University Animal Ethics and Welfare Committee (HZAUFI-2021-0010).

### 2.2. Screening of Surface Fibrin VP56 Epitope of GCRV

The complete sequence of VP56 protein (GenBank accession number: MK675081.1) was analyzed by DNASTAR software, and divided into four antigen fragments and named them: VP56-1, VP56-2, VP56-3 and VP56-4. The VP56 full sequence, four fragments and pGEX-4T-1 vector were digested by restriction enzyme *Bam*H I and *Xho* I (New England Biolabs, Ipswich, MA, USA). The enzyme-digested fragments were purified and ligated together by the T4 ligation enzyme (TakaRa, Tokyo, Japan). The recombinant pGEX-4T-1-VP56-Q, pGEX-4T-1-VP56-1, pGEX-4T-1-VP56-2, pGEX-4T-1-VP56-3 and pGEX-4T-1-VP56-4 plasmids were transferred into *E. coli* strain BL21(DE3), respectively. Positive clones were inoculated with LB medium containing 100 μg/mL ampicillin overnight at 37 °C until the optical density at 600 nm (OD600) reached 0.6. The recombinant VP56 (rVP56) protein was induced to express by adding isopropyl-β-d-thiogalactoside (IPTG, Sigma, St. Louis, MO, USA) into the medium with a final concentration of 1 mM. The induction of rVP56 protein lasted for 6 h at 37 °C. The cells were harvested by centrifugation at 8000× *g* for 30 min at 4 °C.

The bacteria induced by IPTG were collected and re-suspended in PBS (pH = 8.0) and then was broken by a high-pressure crusher in an ice bath for 15 min. The cell mixture was centrifuged at 8000× *g* for 10 min to collect the precipitate. Five target proteins were purified according to the method described previously. The purified protein was identified by SDS polyacrylamide gel electrophoresis (SDS-PAGE). The protein concentration was detected by measuring the absorption at 562 nm by BCA protein assay kit (Solarbio, Beijing, China) with BSA as a standard protein.

After that, the five purified proteins were injected into the abdominal cavity of grass carp, and the control was injected with PBS solution (5 fish in each group). Twenty-eight days later, grass carp were anesthetized with eugenol and the blood was collected from the tail vein. The blood was placed at room temperature for 1 h, centrifuged at 4500 rpm 4 °C for 15 min. Serum samples were collected and stored at −80 °C.

The serum neutralization titers were measured by GCRV-II and CIK cells according to the previous report [39]. Briefly, a serial 1:2 dilution of the serum was mixed with an equal volume of GCRV-II containing 50% TCID50/mL and incubated at 37 °C for 1 h (the dilution range is between 1:2 to 1:40). We added 100 µL of the diluted serum mixture to each well containing CIK cells and incubated the plate at 28 °C for 1 h. Then, the mixtures were gently aspirated, and 0.2 mL of fresh DMEM supplemented with 2% FBS was added back to each well. The 96-well microplates were incubated at 28 °C for 5–7 days. PBS and negative serum were used as negative controls. The disappearance of lesions in each well was regarded as a positive indication of neutralization, and neutralization titers were calculated as the reciprocal highest serum dilution.

### 2.3. Construction of Recombinant Plasmid and Recombinant B. subtilis

The pHT304-CotC-VP56-2 recombinant plasmid was constructed in accordance with the steps as the schematic representation shown below.

Plasmid pHT304 was linearized by using *Bam*H I and *Kpn* I as restriction sites. The amplified products and enzyme-digested products were determined by DNA gel electrophoresis and then recovered by a high-quality gel recovery kit (TIANGEN, Beijing, China). Then, the VP56-2 sequence was inserted into it and ligated with T4 ligase.

We searched the surface protein CotC sequence of *B. subtilis* 168 strain from NCBI and designed primers. *B. subtilis* WB600 is modified on the basis of strain 168. We extracted the DNA of *B. subtilis* WB600 with a bacterial DNA extraction kit and amplified CotC with specific primers (Appendix A). The CotC fragment was ligated to the recombinant plasmid pHT304-VP56-2 by one-step cloning method and transformed into *E. coli* DH5α. The recombinant bacterial solution is spread on an LB medium. After culturing overnight, the monoclonal colonies were picked for positive colony screening, and the positive bacterial solution was sent for sequencing. The recombinant plasmid was extracted with the kit. Then, the recombinant plasmid was electrotransformed into strains WB600, and the recombinant strains were subjected to PCR amplification and DNA sequencing. It was confirmed that the recombinant plasmid was electrotransformed into *B. subtilis* WB600 strains, and the recombinant plasmid pHT304-CotC was electrotransformed as subsequent experimental control (*B. s*-CotC).

### 2.4. Screening Recombinant Strains and Obtaining Recombinant Spores

The method for obtaining recombinant spores is as in previous reports [40,41]. Firstly, the obtained recombinant *B. subtilis* WB600 and *B. s*-CotC were cultured with LB medium, and then the spores were cultured with Difco Sporulation Medium at a ratio of 1:100. One milliliter of the induced spore solution was extracted at different time points (0 h, 6 h, 12 h, 24 h) and centrifuged separately, and we performed SDS-PAGE gel electrophoresis and compared the expression of two recombinant *B. subtilis*. Then, we selected the best expression effect for the subsequent experiments. We selected the recombinant *B. subtilis* with a high expression level to optimize the expression time, extracted 1 mL of induced spore solution at different time points, compared with SDS-PAGE gel electrophoresis experiment and obtained the best induction time for acquiring spores.

The recombinant *B. subtilis* was cultivated. After 48 h, 1 mL of bacterial liquid was taken and centrifuged to collect the spores and treated with 4 mg/mL lysozyme for 30 min at room temperature (RT) to break the remaining sporangia cells. Then, we used 1 M NaCl and 1 M KCl with 1 mM phenyl methyl sulfonyl fluoride (PMSF, Sigma-Aldrich) to wash inhibit proteolysis and then we washed three times with deionized water. After washing, the spores were resuspended in deionized water and treated in a water bath at 65 °C for 1 h in order to kill the remaining WB600 propagules. Finally, we used the obtained spores immediately or stored at 4 °C until use.

### 2.5. SDS-PAGE and Western Blot (WB)

We used SDS-PAGE and WB to examine the expression of VP56-2 on the surface of the spores. First, the surface protein of *B. subtilis* was extracted, and 500 mL of DSM medium was used to obtain the spores of *B. s*-CotC-VP56-2 and *B. s*-CotC. The spore outer protein extraction buffer sodium dodecyl sulfate (SDS)-dithiothreitol (DTT) (0.5% SDS, 0.1 M DTT, 0.1 M NaCl) was treated at 37 °C for 2 h. Then, it was washed 6 times with 1 M Tris-HCl buffer (pH = 8.0) and then suspended in 5 mL of broken buffer (50 mM Tris-HCl, 0.5 mM EDTA, 1 mM PMSF), ultrasonic break for 5 min. After centrifugation, the spore outer protein was collected from the precipitate. Then, we performed 12% SDS-PAGE gel electrophoresis, Coomassie Brilliant Blue R250 stains for 2 h and, finally, observed the expression of the fusion protein in *B. subtilis* and its localization in spores.

We performed SDS-PAGE, transferred the gel to the PVDF membrane and performed the Western Blot experiment. We used fresh 3% skimmed milk powder dissolved in TBST buffer and blocked the membrane for 1 h at 4 °C. The membrane was incubated with mouse anti-rVP56 serum (preserved in our laboratory, diluted 1:1000 with TBST) for 2 h and then washed 4 times with TBST for 10 min each time. Then the membrane was covered with HRP-conjugated goat anti-mouse IgG (1:5000), incubated for 1 h, washed 4 times with TBST and, finally, displayed by the enhanced chemiluminescence (ECL) method.

### 2.6. Immunofluorescence

Using the indirect immunofluorescence method, based on the previous reports [42,43], we detected whether VP56-2 was displayed on the surface of the spore. Briefly, firstly, we fixed 200 μL of the obtained *B. s*-CotC-VP56-2 and *B. s*-CotC spore suspension on a glass slide. Then, the slide with normal goat serum was incubated overnight at 4 °C. Next, we used mouse anti-rVP56 serum (1:200 dilution) as the primary antibody, incubated the slide for 2 h at room temperature and used negative mouse serum as a control. After washing thoroughly, cyanine3 (Cy3)-labeled goat anti-rabbit IgG (1:500 diluted with PBST, Invitrogen, Carlsbad, California, USA) was used as the secondary antibody for incubation for 1 h at RT (in the dark). The samples were incubated for 3–5 min with a DNA staining solution-4’, 6-diamidino-2-phenylindole (DAPI). Finally, the images were captured under a fluorescent microscope (Leica DFC500 Digital Camera, Barnack, Germany) in the dark.

### 2.7. Flow Cytometry

To detect the expression rate of VP56-2 on the surface of spores, flow cytometry was reported previously [44]. In short, approximately 10^6^ CotC-VP56-2 or CotC (used as a control) spores were resuspended in 30 mM NaPO4 buffer (pH = 7.4, containing 2.4% paraformaldehyde and 0.04% glutaraldehyde) at RT for 10 min, then placed on ice for 50 min. After being washed 3 times with PBS, 200 μL of GTE buffer was added to resuspend the pellet, then placed at 37 °C for 30 min. After washing, the spores and mouse anti-rVP56 serum (diluted with 1% BSA-PBS at 1:200) were incubated at 37 °C for 1 h, and the same diluted mouse negative serum was used as a control. The test tube was washed 4 times with PBS. Next, fluorescein (FITC) conjugated goat anti-mouse IgG (1:50 dilution, Akron, OH, USA) was added as a secondary antibody for 1 h at RT. After washing, the spores were resuspended in PBS and transferred to a flow tube and counted with a flow cytometer (BDFACS version). The data were analyzed using FlowJo version 7.6.1 (Tree Star Company, London, UK) software.

### 2.8. Preparation of Feed and Oral Immunization

The *B. s*-CotC-VP56-2 spores and *B. s*-CotC spores were harvested according to the above method. The number of spores calculated by the gradient dilution method was 1.0 × 10^12^ spores/mL. The spores diluted with sterile water were sprayed on the surface of the feed in a spray tank of 1.0 × 10^11^ spores/fish (the optimal concentration is obtained through the preliminary experiment), and then the cod liver oil was wrapped in the mixed feed to prevent the loss of spores in the water. After thorough drying, the feed was stored at −20 °C until use. The feed formula is in Appendix A. The feed formula was made according to the feed formula of Haida Company.

Eight groups of grass carp were divided into four kinds of experimental groups. There were 100 fish in each experimental group; 50 of them were used to measure mortality, and the remaining 50 were used for sampling. Four experimental groups were fed with *B. s*-CotC-VP56-2 spores, *B. s*-CotC spores, *Bacillus subtilis* spores and a normal diet. Grass carp was continuously fed the prepared feed according to 1% of the fish weight (the final concentration is equivalent to 1.01 × 10^11^ spores/g in the *B. s*-CotC-VP56-2 group, *B. s*-CotC group and *B. subtilis* spore group) for 4 weeks. At the same time, the other two groups of grass carp carried out the same feeding activities. The vaccine was administered on the first three days of the week, and normal feed was administered on the last four days of the week. We used this feeding method for four weeks [30].

### 2.9. Sample Collection and Antibody Neutralization Test

Samples were collected in the first, second, third and fourth weeks after the first oral administration. We added clove oil before dissection to soak the anesthetized fish. We obtained blood, spleen, head kidney and intestinal tissue from 5 fish in each group. We put the blood into RT for 1-2 h to clot, stored it at 4 °C overnight and then centrifuged it at 4 °C at 4000 rpm for 20 min to collect the serum. We separated the same length of intestine from each fish, then washed the sample with 1 mL of sterile PBS and used tweezers to scrape off lavage fluid. The lavage fluid was centrifuged at 5000 rpm for 20 min, and then the supernatant was collected as intestinal mucus. Serum and intestinal mucus samples were stored at −20 °C until use. We separated the spleen, head kidney and hindgut tissue, added a 200 μL sample protector (TaKaRaBio, Omatsu, Japan) and stored the samples at −80 °C. The serum biochemical indexes of complement 3 (C3), lysozyme and total superoxide dismutase (TSOD) were assayed by the corresponding commercial kits (Nanjing Jiancheng Bioengineering Institute, Nanjing, China).

Each group’s serum was heat-inactivated at 50 °C for 30 min and diluted at a rate of 1:20 in PBS. Sera were harvested from each sample and used in serum neutralization assays (SN) for the detection of specific antibodies against GCRV II. The SN titer was measured by GCRV II and CIK cells according to the method of Huo et al. [39]. Briefly, a serial 1:2 dilution of the serum was mixed with an equal volume of GCRV II containing 50% TCID50 mL^−1^ and incubated at 37 °C for 1 h (the dilution range is between 1:2 to 1:40). We added 100 μL of the diluted serum mixture to each well containing CIK cells and incubated the plate at 28 °C for 1 h. Then, the mixtures were gently aspirated, and 0.2 mL of fresh M199 supplemented with 2% FBS was added back to each well. The 96-well microplates were incubated at 28 °C for 5–7 days.

### 2.10. Intestinal Colonization Test of Spores

From the first oral immunization to the 6th week, the intestines (2 cm) of 5 grass carp immunized with *B. s*-CotC-VP56-2 spores were taken every week, and the intestinal contents were scraped off under aseptic conditions, then incubated with 500 μL of sterile physiological saltwater at 68 °C for 15 min to kill bacteria except spores. 100 μL of the mixture were spread onto the agar plate and incubated at 37 °C for 14–18 h. We selected 10 colonies from each fish’s plate and mixed them together for PCR amplification to detect the colonization of spores in the intestinal tract. Finally, the amplified products were identified by 1% agarose gel electrophoresis.

### 2.11. Virus Challenge and Viral Load Determination

After 4 weeks of oral administration, grass carp (*n* = 50 per group) were injected intraperitoneally with 1 × 10^6^ TCID50 GCRV 097. The GCRV 097 strain was stored in our laboratory. Each tank had a heating rod to keep the water temperature stable at 28 °C. We monitored clinical signs and mortality. The important immune tissues-head kidney and spleen were collected at the indicated time points. No fish died in all the groups 9 days post challenge.

The cumulative survival rate was recorded to evaluate the protective effect of CotC-VP56-2 spores. The relative survival rate (RPS) was calculated according to Amend’s method. The calculation formula: RPS = (1 − (mortality rate of immunization group/mortality rate of control group)) × 100. In addition, qRT-PCR was used to detect the viral load in the spleens and head kidneys of 5 dead fish in each group to confirm that the fish died of GCRV infection and to evaluate the antiviral effect of the oral vaccine.

### 2.12. Determination of Immune-Related Genes Expression by qRT-PCR

Two important immune tissues—spleen and head kidney—were collected, and mRNA expressions of immune-related genes (*IL-1β*, *TNF-α*, *IFN1* and *MHC-II*) were investigated after oral administration by qRT-PCR. In short, total RNAs were extracted with TRIzol reagent, then reverse transcribed into cDNA with the cDNA synthesis complex kit. qRT-PCR was performed. The primers were shown in Appendix A. qRT-PCR was carried out by a CFX96 real-time PCR detection system (Bio-Rad, Hangzhou, China) in a final volume of 20 μL. For internal control, 18S rRNA was used, and the relative expression was calculated by the 2ΔΔCt method [45].

### 2.13. Histopathological Examination

The spleen and trunk kidney were immediately fixed in 10% neutral formalin for 24 h, dehydrated, embedded in paraffin and sectioned. A four-micrometer section of the sample was mounted on a glass slide coated with (3-Aminopropyl) triethoxysilane. After deparaffinization, the sections were rehydrated in xylene, stained with hematoxylin and eosin (HE) and mounted with neutral glue. Finally, we captured these images.

### 2.14. Statistical Analysis

The results were presented as means ± standard deviation (SD). Multiple pairwise comparisons were performed using one-way analysis of variance (ANOVA) with Bonferroni tests using GraphPad Prism vision (version 6.01 for Windows) and Image J (1.8.0). The experimental data of each group were subjected to a Kruskal–Wallis test, followed by Dunn’s multiple comparison test (with Bonferroni adjustment). *p* < 0.05 was considered a statistically significant difference. We used Image J software to calculate the ratio of hemosiderin agglutination, hypertrophied nuclei area and vacuolization area in spleen and trunk kidney tissue sections.

## 3. Results

### 3.1. Expression of VP56 Protein and Screening of Epitopes

The full length of the coding sequence of the VP56 protein is 1539 bp. We analyzed the epitope region using the DNASTAR tool and divided it into four fragments. They were named VP56-1, VP56-2, VP56-3 and VP56-4. These four fragments and the full length of VP56 were amplified and ligated to the pGEX-4T-1 plasmid. The recombinant plasmids were transformed into the *E. coli* expression strain BL21(DE3) and successfully expressed the target protein (Figure 1A). ELISA showed that the VP56-2 fragment bound significantly better to the antibody than the other fragment groups and slightly better than the full-length group (Figure 1B). CIK cells were incubated with a mixture of GCRV-II and antiserum for neutralizing antibody assays (Figure 1C). Our results showed that all cells in the GCRV-II group died, while all cells in the PBS group survived. The VP56-2 serum group had the strongest ability to neutralize the antibody, significantly stronger than the other groups. Cells in the anti-VP56-2 serum group only started to die at a serum dilution ratio of 1:10,000. We used Image J software to analyze the crystalline violet results in greyscale values (Figure 1D). There was a significant difference between the serum group with the anti-VP56-2 fragment and the serum group with the anti-VP56-4 fragment. The CIK cells had a better immune effect when infected with CIK cells after incubation with the anti-VP56-2 fragment serum group and GCRV-II. Based on this result, it suggests that the VP56-2 fragment has good antibody binding ability and is a potential epitope for GCRV-II.

### 3.2. Successful Construction of pHT304-CotC-VP56-2 Recombinant Plasmid and B. s-CotC-VP56-2 Recombinant Spores

The VP56-2 fragment was 344 bp in length. The CotC fragment was ligated to pHT304-VP56-2 using the One-Step PCR Cloning method, and the plasmid was used as a template for recombination (Figure 2A). The recombinant plasmid spots were transformed into *Bacillus subtilis* WB600, sporulation was induced in DSM medium, and CotC spores were harvested using the same method as the control method. Protein expression was detected by SDS-PAGE gel electrophoresis. The molecular weight of the CotC-VP56 fusion protein was approximately 26.8 kDa, which is comparable to that of VP56-2 (18 kDa) and CotC (8.8 kDa). As shown, the CotC-VP56-2 fusion protein was expressed in recombinant spores of *B. subtilis* strain WB600, but no matching bands were observed in spores of CotC (Figure 2B). Expression of the CotC-VP56-2 fusion protein increased progressively with increasing incubation time (from 0 h to 72 h) (Figure 2C). In addition, Western Blot analysis showed that the extract precipitates containing the coat protein of the spore contained significant amounts of CotC-VP56-2 fusion protein, but there were few corresponding bands in the supernatant (Figure 2D). These results suggest that the pHT304-CotC-VP56-2 recombinant plasmid and recombinant spores expressing the CotC-VP56-2 fusion protein have been successfully constructed.

### 3.3. Immunofluorescence and Flow Cytometry Analysis of VP56-2 Expressed on the Surface of Spores

After successfully constructing the pHT304-CotC-VP56-2 recombinant plasmid and the CotC-VP56-2 spore expressing the CotC-VP56-2 fusion protein was successfully constructed, we conducted immunofluorescence and flow cytometry experiments to further prove that the VP56-2 protein was expressed on the surface of the spores. The results of immunofluorescence experiments showed that red fluorescence (VP56-2) was widely distributed on the surface of CotC-VP56-2 spores incubated with mouse anti-rVP56 serum (Figure 3A), while the surface of CotC-VP56-2 spores cultured with negative mouse serum has no fluorescence (Figure 3B). The results of flow cytometry showed that VP56-2 was distributed on the surface of CotC-VP56-2 spores, and the positive rate (82.22%) was high. The positive rate of CotC spores incubated with mouse anti-rVP56 serum was low (2.15%), or the positive rate of CotC spores incubated with mouse negative serum (0.06%) was low. The results show that the VP56-2 fragment is expressed on the surface of *B. subtilis*.

### 3.4. Colonization of Oral Spores in the Intestine and Evaluation of the Protective Effect of CotC-VP56-2 Spores

Weekly intestinal samples were collected for 6 weeks after feeding to test for spore colonization. Spores could be detected in the gut in the first week after feeding (Appendix A). Specific PCR products (approximately 344 bp by agarose gel electrophoresis) were amplified with specific primers using plate colonies as a template, indicating that CotC-VP56-2 spores could colonize the gut of grass carp after oral immunization. A greyscale value analysis of the PCR-amplified bands using Image J software revealed that the spores colonized the gut more with an increased feeding time (Figure 4B), peaking at week 4 and remaining largely unchanged in weeks 5 and 6. Therefore, we chose to carry out subsequent challenge experiments after 4 weeks of feeding.

In the vaccination and viral infection experiments, the expression of *IgM* mRNA for antibodies in the *B. s*-CotC-VP56-2 group 3 dpi (spleen) or 7 dpi (head kidney) increased significantly, with the highest expression in all four groups (Figure 4D,E). The trend in IgM expression was consistent with the results of the greyscale analysis of spore colonization. Afterwards, we recorded the mortality of each group of experimental fish after the challenge. On the third day, grass carp began to die. On the eighth day, all of the control group died. In total, 56% of grass carp in the *B. s*-CotC-VP56w-2 group survived, 21% of grass carp in the *B. s*-CotC group were protected and 18% of grass carp in the *B. subtilis* group survived (Figure 4C). The RPS of *B. s*-CotC-VP56-2 was 56%. The RPSs of each group are shown in Appendix A. In summary, feeding *B. s*-CotC-VP56-2 stimulated the immune response of the organism, increased specific IgM and provided some degree of protection against grass carp hemorrhagic disease.

The serial dilution sera were mixed with GCRV II and incubated in CIK cells for neutralizing the antibody titer assays. Our results showed that neutralized antibody titers were significantly high in *B. s*-CotC-VP56. Before the challenge, each group’s score was low but slowly rose after the challenge. (Figure 5D).

### 3.5. Detection of Serum Innate Immunity Level

In the assay of serum innate immune markers, complement C3 levels in the CotC-VP56-2 group improved significantly after 3 days, with C3 concentrations returning to normal after 28 days (Figure 5A). After 30 days, there was a rapid increase, which was statistically different compared to the other groups. Meanwhile, TSOD activity increased only in the *B. s*-CotC-VP56-2 group and increased more rapidly (Figure 5B). After feeding, Lysozyme activity increased in the *B. s*-CotC-VP56-2 group compared to the other groups, then decreased slightly and increased further after tapping, with a significant difference compared to the other groups at 32 days (Figure 5C). In general, feeding recombinant *Bacillus subtilis* the VP56-2 oral vaccine was effective in increasing serum innate immunity levels.

### 3.6. Expression of Immune-Related Genes in Different Tissues

The mRNA expression of representative immune genes, including *IL-1β*, *TNF-α*, *IFN1* and *MHC-II*, was measured by qRT-PCR in the spleen and head kidney at different time points after vaccination and challenge (Figure 6). The mRNA expression of immune genes in the head kidney and spleen was not significantly up-regulated in the *Bacillus subtilis* group, the *B. s*-CotC group and the control group. Immunogens were significantly up-regulated in the vaccine group. mRNA expression of *IL-1β* and *TNF-α* was significantly increased in the *Bacillus subtilis*, *B. s*-CotC and vaccine groups on the day after the challenge, while the *B. s*-CotC-VP56-2 group was most significantly up-regulated and differed significantly from the other groups (Figure 6A–D). On the third day after the challenge, the mRNA expression of *MHC-II* and *IFN1* in *Bacillus subtilis*, *B. s*-CotC and *B. s*-CotC-VP56-2 were significantly up-regulated, and the *B. s*-CotC-VP56-2 group was significantly different from the other groups (Figure 6E–H). With the change in *IL-1β* expression, the mRNA expression of *IFN1* and *MHC-II* subsequently increased and decreased significantly. These results suggest that the oral administration of *B. s*-CotC-VP56-2 for the prevention of GCRV enhances innate and adaptive immunity in grass carp.

### 3.7. B. s-CotC-VP56-2 Oral Vaccine Can Effectively Alleviate Viral Infections

After we evaluated the protective effect of *B. s*-CotC-VP56-2 spores, we verified whether the protective effect of oral administration of *B. s*-CotC-VP56-2 spores was achieved by limiting virus replication in grass crap. Following the GCRV-II challenge, GCRV transcript levels were quantified by qRT-PCR in the spleen and trunk kidney. In both organs, GCRV VP4 mRNA expression levels were lower in grass carp immunized with oral *B. s*-CotC-VP56-2 than in non-immunized fish. In addition, the spleen had the highest viral load of all tissues in unimmunized grass carp. In contrast, grass carp immunized with oral *B. s*-CotC-VP56-2 spores had significantly lower viral loads in the spleen and trunk kidney (Figure 7K,L). This suggests that oral *B. s*-CotC-VP56-2 immunization effectively limited the replication of GCRV in grass carp.

Then, to assess the extent of tissue damage in each group, we performed HE staining of spleen and trunk kidney tissues one week after onset (Figure 7A–J). We used healthy grass carp as control. Severe lesions, including cavitation, hyperplasia nuclear and hemosiderin agglutination, were evident in the control, *B. subtilis* and *B. s*-CotC groups but not so much in the *B. s*-CotC-VP56-2 group. There was significant protection in the *B. s*-CotC-VP56-2 vaccine group compared to healthy tissue. These results suggest that *B. s*-CotC-VP56-2 treatment is effective in protecting fish tissue from damage. This is consistent with the results of the viral load assay. Detailed statistical results of tissue lesions were presented in Appendix A.

In summary, oral administration of the *B. s*-CotC-VP56-2 vaccine was effective in protecting fish tissues from GCRV damage and limiting GCRV replication and proliferation in vivo.

## 4. Discussion

The production of farmed grass carp in China has steadily increased, but the development of GCRV vaccines has been slow, and most of the developed vaccines have not been commercialized. The first vaccine for grass carp hemorrhagic disease was an “inactivated tissue vaccine” developed in the 1960s [46]. Currently, the most common method of preventing GCRV is immunization with inactivated virus vaccines prepared by research teams at research institutions or universities [47]. However, inactivated vaccines have many drawbacks. In contrast, subunit vaccines based on genetic engineering techniques are produced faster, more effective and stronger. Therefore, subunit vaccines are more suitable for large-scale production and application. In this study, a strong immune fragment of VP56 was first screened, and a VP56-2 fragment was screened to make a subunit vaccine. This experiment selected the high-safety probiotic *B. subtilis* as an oral vector for oral immunization, and the vaccine was evaluated. The results showed that the recombinant *Bacillus subtilis* VP56-2 oral vaccine was effective in producing an immune response in grass carp, and although the protection rate was relatively low compared to the intraperitoneal injection of the subunit vaccine VP56, the amount of injected protein was greater than the amount of oral protein, so the oral recombinant *Bacillus subtilis* VP56 vaccine may be a candidate vaccine against GCRV.

The VP56 protein of the GCRV-II virus is unique. There is no similar protein in GCRV-I, while VP55 in GCRV-III has only 15% similarity [5]. VP56 is predicted to be fibrin, and fibrin in mammalian virus σ1 is involved in cell attachment during virus invasion [48,49]. Recent studies have shown that VP56 binds to fish cells when GCRV-II enters the fish body. This shows that after the virus enters the fish body, VP56 can act as a cell attachment protein [50,51]. Studies have shown that VP56 negatively regulates IRF7 through the ubiquitination-proteasome degradation pathway, which is a new way for GCRV immune escape [16]. Therefore, this experiment uses VP56 protein to prepare oral subunit vaccines.

Existing studies have found that the screening of antigenic epitopes can significantly improve the body’s immune function to suppress the virus. Recent studies have found that the use of super sf9 cells to produce the n-terminal fragment system of Pfs230 in baculovirus expression can be very effective. It is good to induce the mouse immune system to produce antibodies that inhibit Malaria [52]. In addition, another team divided the VP4 protein of the GCRV-II virus into four fragments, used ELISA experiments to screen the epitope, screened out the strongest binding ability of VP4-3 to the antibody and, finally, made a nano-vaccine, which can effectively resist the GCRV of the fish [30]. Therefore, this experiment was also designed to split the VP56 protein into four fragments. Through ELISA experiments, we found that the binding ability of VP56-2 and VP56-4 antibodies was significantly higher than that of VP56 full-length and other fragments. The results of the neutralizing antibody experiments on VP56-4 and VP56-2 showed that the cells of VP56-4 started to die in large numbers when the serum dilution ratio was 1:1000, indicating that the neutralizing ability of the VP56-4 serum antibody started to decrease. At 1:5000, VP56-2 started to die in large numbers. Meanwhile, Image J analysis of crystalline violet was performed, and it was found that there was a significant difference between the VP56-2 group and the VP56-4 group, and the cell survival effect was significantly better than that of the VP56-4 group. Based on these results, it can be concluded that VP56-2 is a potential antigen site with a high antibody binding capacity. Later, VP56-2 was fused to the *Bacillus subtilis* exocyst protein CotC [53], and anchored on the surface of *B. subtilis* to make a recombinant *B. subtilis* oral vaccine. After 4 weeks of administration, a GCRV-II stimulation assay was performed. The results of the protection rate after the challenge showed that the control group all died on day 7, while the survival rate of the *B. s*-CotC-VP56-2 group was 52%. Thus, oral administration of recombinant *Bacillus subtilis* VP56-2 could improve the ability to resist GCRV-II.

Antibodies play a key role in the humoral immune response. Among the three subtypes of immunoglobulin (Ig), *IgM* is widely present in the circulation of the fish system [54]. In our study, upregulated expression of *IgM* mRNA increased after oral vaccine administration. expression of *IgM* mRNA decreased on the first day after the challenge and then increased rapidly. The rapid upregulation of *IgM* mRNA suggested the memory B cells can be activated rapidly [55]. Moreover, *IgM* expression increased rapidly after the challenge, with the oral vaccine group being higher than the other groups. In our results, *B. s*-CotC-VP56-2 treatment could induce strong immune responses and high titers of neutralizing antibodies post-vaccination and challenge. After incubating the virus with the serum of the *B. s*-CotC-VP56-2 group, the state of CIK cells was much better than other groups, indicating that the serum of *B. s*-CotC-VP56-2 group had a good neutralization effect on GCRV II. The results of the spore colonization experiment were also consistent with changes in *IgM*. This result suggests that oral administration of *B. s*-CotC-VP56-2 to grass carp elicits a strong humoral response and that the antibody provides continuous protection against GCRV infection in the grass crap.

The serum innate immunity index usually reflects the level of immunity of the animal. Previous studies have shown that resistance of grouper to bacterial and viral pathogens is associated with increased complement C3, TSOD and lysozyme activity [56]. The determination of serum biochemical indexes has been widely used in the clinical diagnosis of fish physiology to determine the general state of health [57]. Serum biochemical indices can reveal the response of fish to changes in internal and external factors.

The complement system is one of the major components of non-specific humoral immunity, and it plays an important role in labeling and clearing exogenous microorganisms, as well as regulating inflammatory and specific immune responses [58]. In addition, the complement system plays an essential role in alerting the host of the presence of potential pathogens. Therefore, the measurement of complement is very important. There are three pathways for the activation of the complement system, and C3 is a key link in three pathways [58]. Therefore, we measured the change of C3 in our study. The level of complement C3 in the serum increased rapidly after vaccine administration and challenge. This result suggested that complement C3 is actively involved in antigen labeling and enhances specific and non-specific immune responses to eliminate antigens.

The total superoxide dismutase (TSOD) is a very important antioxidant enzyme in vivo [56]. It mainly removes superoxide from animal body fluids or tissues and plays a role in reducing or removing superoxide oxidative damage from animal cell membranes or reducing active components in cells. Studies have shown that the activity of TSOD is closely related to the immune level of the body. Lysozyme is mainly produced by monocytes and granulocytes, which can not only destroy and remove invaders but also improve the digestive function of macrophages and enhance immunity.

Superoxide dismutase and lysozyme are also closely associated with the level of immunity in fish [59]. TSOD and lysozyme showed an increasing trend after vaccine feeding and challenge. The results suggest that *B. s*-CotC-VP56-2 can induce beneficial immune responses in the blood.

It is necessary to study the mechanism of action of vaccines on innate immunity and adaptive immunity. We used qRT-PCR to detect the expression of four immune-related genes in spleen and head kidney tissues 4 weeks after the first oral administration. *IL-1β* is an important early inflammatory cytokine, which plays an important role in regulating defensive and pathological innate immune responses [60]. In cells, *TNF-α* and *IL-1β* are two effective pro-inflammatory cytokines, which play an important role in immunity, inflammation, and control of cell proliferation, differentiation and apoptosis [61]. This study found that *IL-1β* and *TNF-α* increased rapidly after 4 weeks of challenge with an oral vaccine, which shows that this treatment method can better improve the early inflammatory response. This indicates that monocytes and macrophages may have a strong inflammatory response to antigens entering fish, which is necessary for enhancing fish’s immunity and ultimately eliminating foreign pathogens.

The *IFN* response plays an important role in protecting the host from viral infections [62,63] and bacterial infections [64]. The host has conservative pattern recognition receptors (PRRs), which can sense viral RNA and trigger a variety of intracellular signaling pathways, including the retinoic acid-inducible gene I (*RIG-I*)-like receptor (*RLR*) pathway, which ultimately leads to the production of *IFN*. Establish anti-viral status [65,66,67]. The specific *RLR* pathway is as follows: After binding to viral RNA, the n-terminal cysteine recruitment domain (CARD) of the RLR family (including melanoma differentiation-related gene 5 (*MDA5*) and another CARD protein, mitochondrial antiviral signal protein (MAVS, also known as the interaction of VISA, IPS-1 and Cardif) [68,69,70,71]. Then, the signal is transmitted to IFN regulatory factor 3 (*IRF3*) to activate MITA (also known as STING, ERIS or MYPS) and TANK binding kinase 1 (*TBK1*). Major histocompatibility complex class II (*MHCII*), *MHCII*, plays an important role in the adaptive immune response by presenting antigens to CD4^+^ T lymphocytes [72], and CD4L is a subset of T helper cells and [73] is a key marker molecule, which provides a basis for solving the participation of T cells [74]. Consistent with the immune response, the grass carp taking *B. s* -CotC-VP56-2 rapidly increased *IFN1* and *MHC-II* on the third day, which was significantly higher than the other groups. These results are consistent with the previous results. These results indicate that the oral *B. s*-CotC-VP56-2 vaccine can enhance the immune protection of GCRV by inducing higher levels of adaptive humoral and cellular immunity. In summary, the vaccine can not only induce innate immunity but can also induce humoral and cellular immunity of immunized fish. However, whether specific mucosal immunity in the fish intestine participates in this immune protection remains to be evaluated. In teleost, the immunoglobulin isotype *IgZ* or *IgT* has been shown to be ubiquitous in the intestine, skin, and gills and plays an important role in mucosal immunity. Therefore, the *IgZ* or *IgT* response to the VP56-2 antigen administered by recombinant spores is worth further studying in our future research.

In addition, we investigated the protective effect of the vaccine on the tissues after the challenge. In previous studies, histopathological sections could visualize the pathological damage of GCRV-II infection. Typical manifestations are multi-organ hemorrhage, intestinal atrophy and rupture, and congestion of gill filaments, which affects gas exchange and leads to hypoxia. The vascular wall is damaged, and a large number of red blood cells can be seen in the muscle and intestine [75]. GCRV infection can trigger the caspase process of vascular endothelial cell apoptosis, inducing hemorrhage. Severe vacuolation and cell necrosis caused in the kidney and liver weakened metabolism and immune function, leading to massive spread and proliferation of the virus [76]. The results of our histological analysis showed that the spleen and torso kidney of the *B. s*-CotC-VP56-2 group had the mildest lesions and the control group had the most pronounced lesions; and our results of viral load in each tissue by qRT-PCR also showed that the spleen and torso kidney of the *B. s*-CotC-VP56-2 group had the least viral load *B. s*-CotC-VP56-2 group, which was consistent with histological results, which, together with the analysis of spore colonization, suggest that grass carp can pass the intestinal mucosal barrier. Uptake of *B. s*-CotC-VP56-2 to aid in the clearance of GCRV-II resulted in a reduction in the amount of the virus in the tissues, leading to a reduction in pathological damage in the tissues. These results suggest that spores are not degraded in grass carp and that oral administration of *B. s*-CotC-VP56-2 is effective in protecting tissue function from damage.

## 5. Conclusions

In short, we screened out the potential epitope VP56-2 of GCRV-II and expressed it on the surface of *B. subtilis* using the *B. subtilis* surface expression system to make an oral subunit vaccine. The protection rate experiment found that it can be effectively prevented. GCRV infection. The antiserum against rVP56-2 protein showed a neutralizing effect on GCRV, indicating that VP56 is located on the outer surface of the virus particle, which may be the target of neutralizing antibodies. In our experiments, we also found that spores can not only promote innate immunity by expressing the immune gene VP56-2 but can also promote humoral and cellular immunity after oral immunization. Considering the convenience, safety and protection of the CotC-VP56-2 vaccine, we should use this recombinant oral vaccine as a potential vaccine candidate for the prevention of GCRV. This work also emphasizes that *B. subtilis* spores are a powerful oral vaccine delivery system that can be widely used in other oral vaccines. Future applications are very promising and can be used in the veterinary and medical fields.

## Figures and Tables

**Figure 1 viruses-14-00030-f001:**
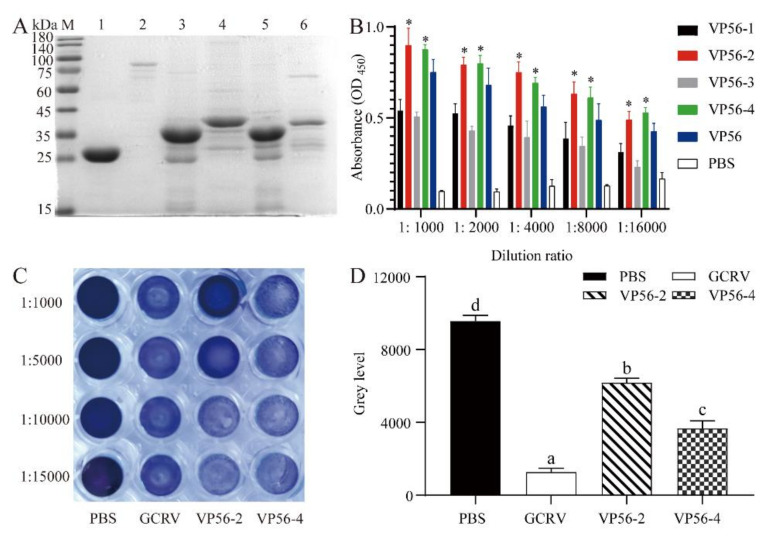
Screening VP56 epitopes. (**A**) SDS-PAGE to detect the full-length and fragment protein expression of VP56. Lane M: protein Marker; Lane1-6: GST as control, VP56 (81 kDa), VP56-1 (32 kD), VP56-2 (43 kD), VP56-3 (31 kD), VP56-4 (42 kD). (**B**) ELISA experiments of full-length and fragment proteins to screen for proteins with immune effects. Data were presented as the means ± SD. Values that were significantly different from the control were indicated by asterisks (One-way ANOVA, * *p* < 0.05). (**C**) Neutralizing antibody test. The serum dilution ratios are 1:1000, 1:5000, 1:10000 and 1:15000. PBS group: no challenge group; GCRV-II group: only challenge group; VP56-2 and VP56-4 groups: antiserum and virus incubation. (**D**) Image J analysis of crystalline violet staining results. Different superscript letters in each group (a–d) denote significant variations suggested by the Kruskal–Wallis statistics at 95% of significance, followed by Dunn’s test with Bonferroni adjustment as the post hoc test (*p* < 0.05).

**Figure 2 viruses-14-00030-f002:**
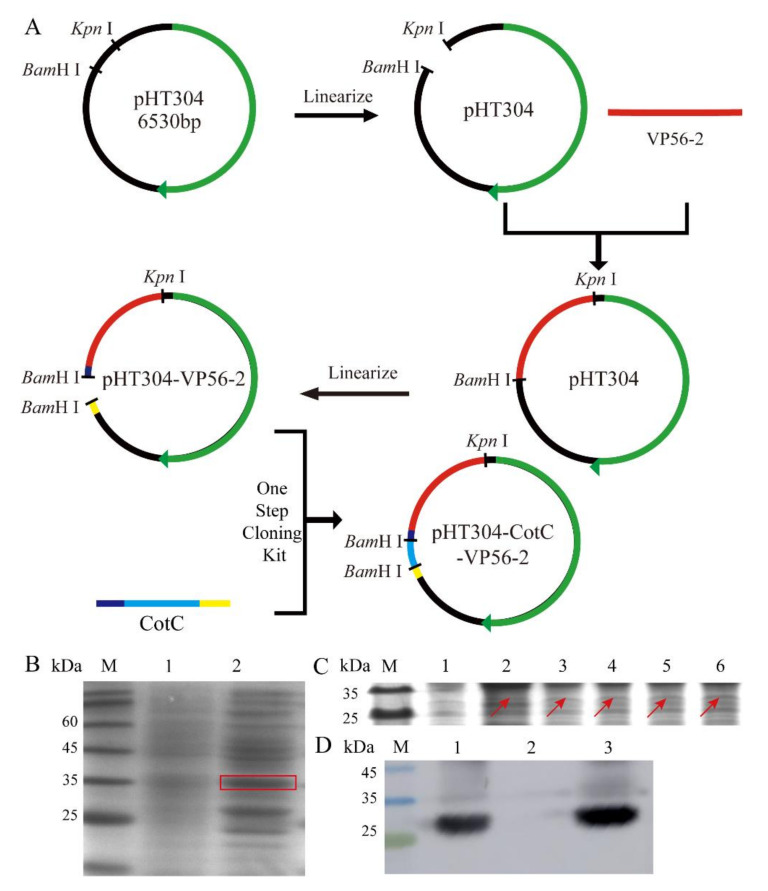
Construction and expression of the pHT304-CotC-VP56-2 recombinant plasmid. (**A**) Schematic diagram of pHT304-CotC-VP56-2 recombinant plasmid construction. (**B**) The recombinant plasmid was expressed in *B. subtilis* WB600. Lane M: protein marker, Lane 1: CotC spores, Lane 2: 48 h induction (**C**) The expression of recombinant *B. subtilis* at different time points. Lane M: protein marker, Lane 1–6: 0 h, 6 h, 12 h, 24 h, 48 h and 72 h induction. (**D**) Western Blot experiment was used to verify the expression of VP56-2. Lane M: protein marker, Lane 1: precipitation after sonication, Lane 2: supernatant after sonication, Lane 3: not sonicated.

**Figure 3 viruses-14-00030-f003:**
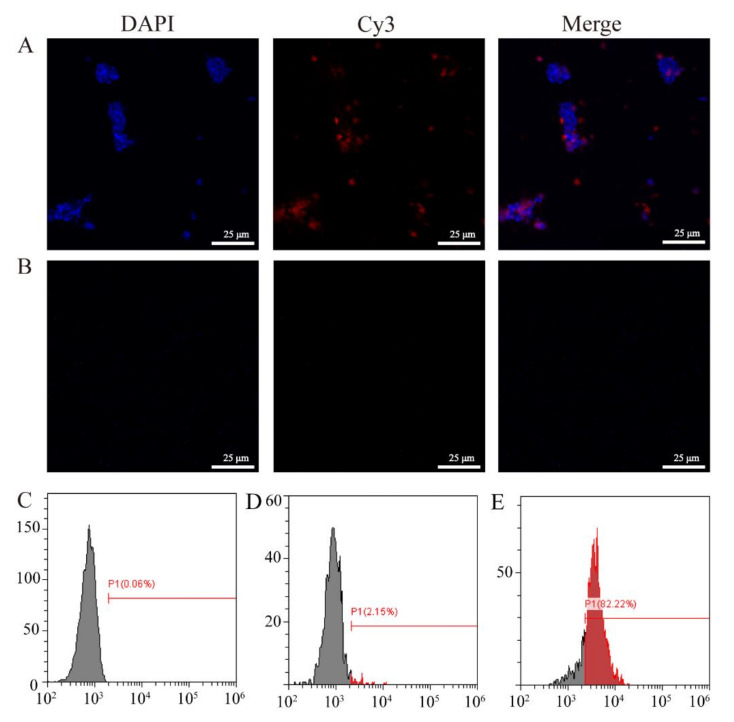
The expression of VP56-2 protein on the surface of spores was verified by immunofluorescence and flow cytometry. (**A**,**B**): The expression of CotC-VP56-2 spores by the immunofluorescence method. CotC-VP56-2 spores were incubated with mouse anti-VP56 serum (**A**), CotC-VP4 spores were incubated with pure mouse serum (**B**) and then with Cy3-labeled goat anti-mouse IgG. The spores of CotC-VP56-2 are stained red, and the nucleus is stained blue. Scale bar: 25 μm. (**D**,**E**): Flow cytometry to detect the positive expression rate of VP56-2 on the surface of recombinant spores. CotC-VP56-2 spores were incubated with pure mouse serum (**C**), CotC spores and CotC-VP56-2 spores were incubated with mouse anti-VP56 serum (**D**,**E**) and then incubated with FITC-conjugated goat anti-mouse IgG.

**Figure 4 viruses-14-00030-f004:**
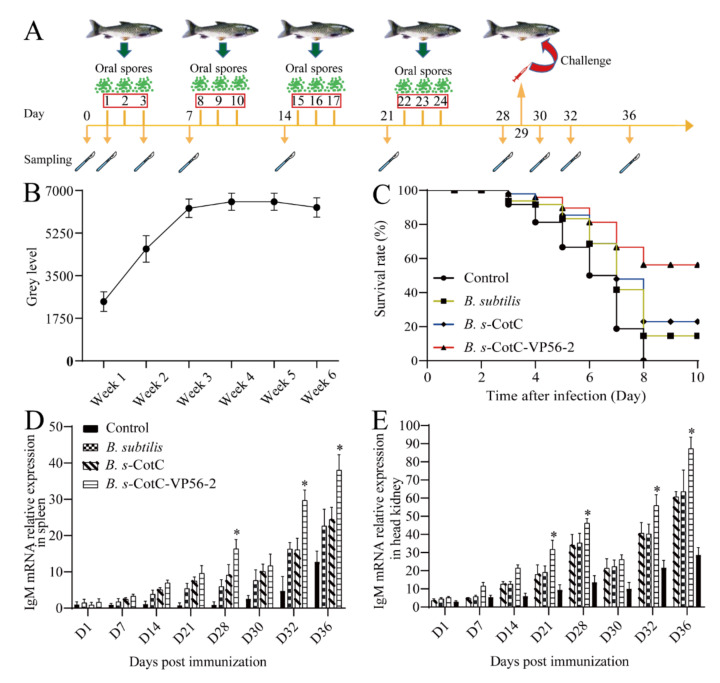
Preparation and in vivo test of oral *B. subtilis* vaccine. (**A**): Flow chart of vaccination, attack and sampling. (**B**): Image J analysis of spore colonization PCR; (**C**): Cumulative survival rate of grass carp inoculated with GCRV-II. On day 0, each group of fish was fed ordinary feed, *B. subtilis* feed, *B. s*-CotC mixed feed and *B. s*-CotC-VP56-2 feed. Each group of fish (n = 50) was injected with 200 μL of GCRV virus on the 29th day. Next time, Next, the death of fish in each group was observed. On day 8, all control fish died. (**D**,**E**): qRT-PCR was used to detect the serum *IgM* expression level of the grass carp’s spleens and head kidneys, and 18S rRNA was used as an internal control gene. Data are expressed as mean ± SD. Compare the *B. s*-CotC-VP56-2 group with the control group and other groups. Values that are significantly different from the control are indicated by asterisks (One-way ANOVA, * *p* < 0.05).

**Figure 5 viruses-14-00030-f005:**
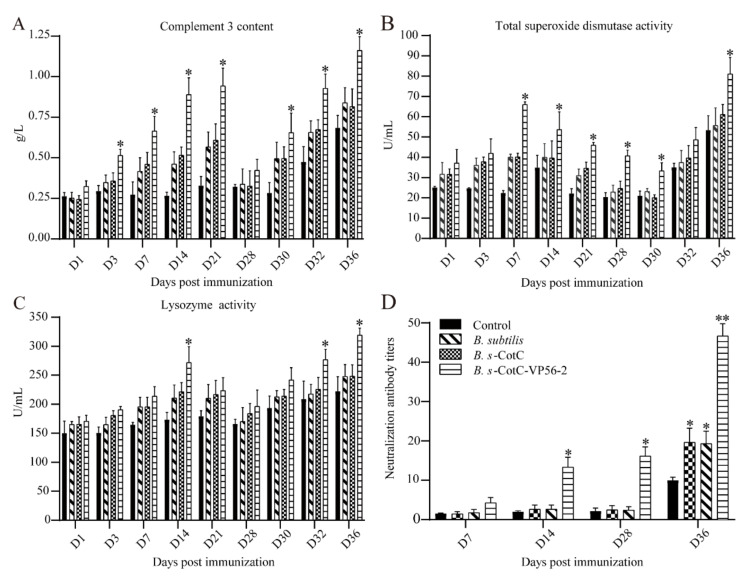
Serum innate immunity index. Complement C3 (**A**), lysozyme (**B**) and total superoxide dismutase (**C**) were measured using commercial kits (China Nanjing Jiancheng Institute of Bioengineering). Data are presented as the means ± SD. Values that were significantly different from the control are indicated by asterisks. The neutralization assay of anti-GCRV II serum in vitro. (**D**) The reciprocal of the highest serum dilution needed to prevent cytopathic disease in each experimental group and at different time-points. Data were presented as means ± SD (*n* = 4). (One-way ANOVA * *p* < 0.05, ** *p* < 0.01).

**Figure 6 viruses-14-00030-f006:**
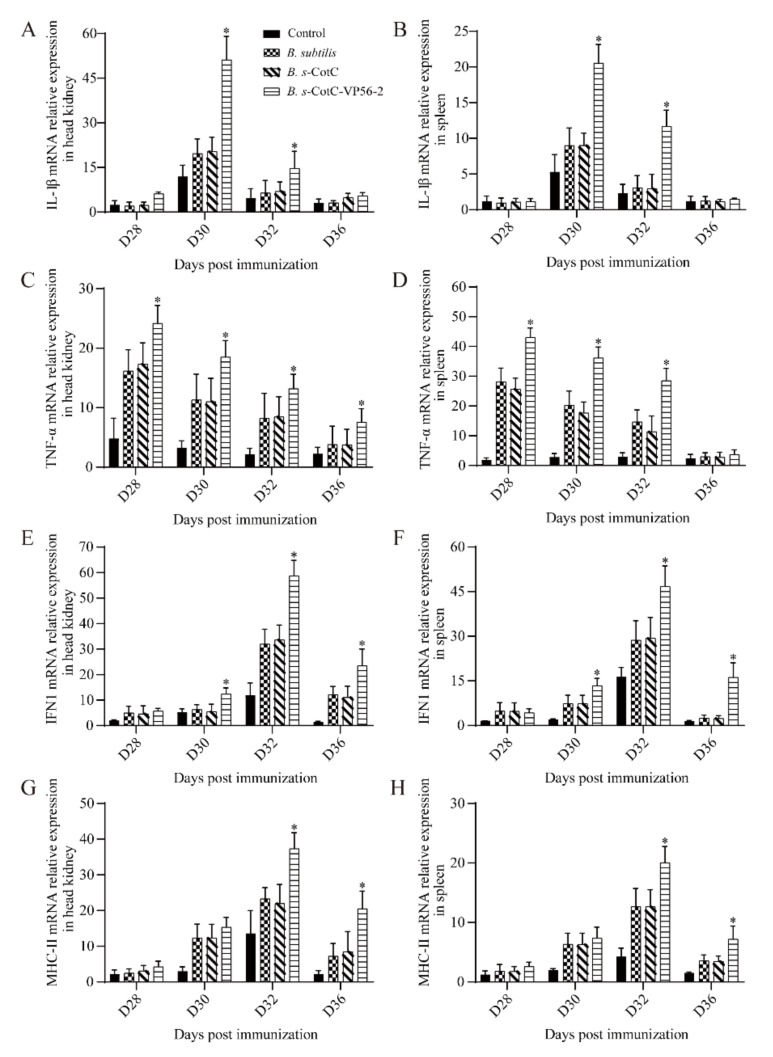
The mRNA expression pattern of representative important immunomodulatory and effector genes in head kidney and spleen tissues. Check *IL-1β* (**A**,**B**), *TNF-α* (**C**,**D**), *IFN1* (**E**,**F**) and *MHC-II* (**G**,**H**) transcripts by qRT-PCR. For an internal reference gene, 18S rRNA was used. Data are expressed as mean ± SD, and values that are significantly different from the control are indicated by asterisks (one-way analysis of variance, * *p* < 0.05).

**Figure 7 viruses-14-00030-f007:**
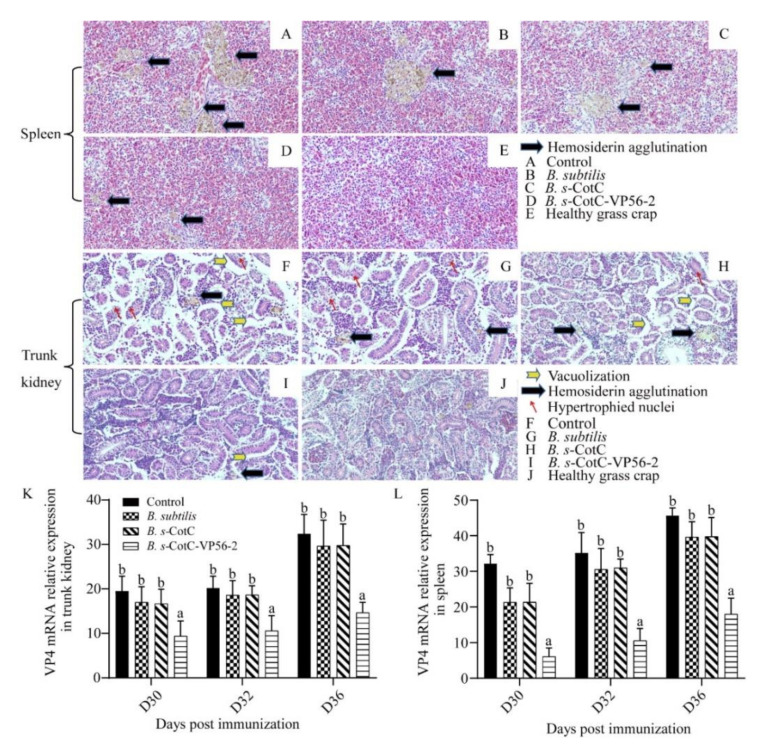
Tissue damage and viral loads in spleen and trunk kidney after challenge. On day 0, grass carp were fed with ordinary feed, *B. subtilis* feed, *B. s*-CotC mixed feed and *B. s*-CotC-VP56-2 feed. GCRV-II was injected intraperitoneally on 29 dpi. Spleen and trunk kidney tissues were collected on 36 dpi, sectioned and stained with HE. Healthy grass carp tissue was used as a blank control. In the spleen, vacuolation and nuclear cysts are the two main signs (**A**–**E**). The main manifestations in the trunk kidney are the formation of vacuoles, the agglutination of hemosiderin and the hypertrophy of the nucleus (**F**–**J**). mRNA expression profile of GCRV-II in trunk kidney (**K**) and spleen (**L**). We used the qRT-PCR method to detect and 18S rRNA as an internal reference gene. Different superscript letters in each group (a,b) denote significant variations suggested by the Kruskal–Wallis statistics at 95% of significance, followed by Dunn’s test with Bonferroni adjustment as the post hoc test (*p* < 0.05).

## Data Availability

Not applicable.

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
