# Peer review of "Oral Administration of Bacillus subtilis Subunit Vaccine Significantly Enhances the Immune Protection of Grass Carp against GCRV-II Infection"

_viruses, 2021, doi:10.3390/v14010030_

Round 1

Reviewer 1 Report

In this manuscript, a subunit vaccine, VP56-2, was presented and exhibited to cell surface by B. subtilis, and its protective effect is good. The subject addressed in this article is worthy of investigation. The design is rational and organization is appropriate. The result is abundant and the conclusions are supported by the data. Overall, the manuscript is meaningful. Several concerns:

Line 378, fig. 4C is an error

Lines 399-404, after infecting three days with virus, the fish began to die, and in 8 days, all of the control fish died. Is it reasonable?

Fig. 5 is unclear.

Writing of B. subtilis-CotC-VP56-2 and B. s-CotC-VP56-2 need to be consistent.

The relative survival rate (RPS) appears in the material and methods, but not in the result.

Many writing errors need be corrected in the whole manuscript.

Reviewer 2 Report

Grass carp reovirus (GCRV) is a severe virus threaten the grass carp culture industry every year. Immune prevention is the most effective way to control this disease. In this study, the authors development an oral administration of bacillus subtilis subunit vaccine which have higher protection rate. Moreover, the oral immunization used in this manuscript was convenience and exhibited application prospects. This work would be important to the GCRV disease control. There are some comments and suggestions for this work before published.

Comments:

  1. In this study, the authors showed that the CotC-VP56-2 spore located in the intestine through in vitro culture. Whether the CotC-VP56-2 spore recovered in the intestine and displayed the VP56-2 fragment? The VP56-2 mRNA and the proteins should be detected in the intestine after oral immunization.
  2. The serum neutralization titers were an important indicator to prove this vaccine after oral immunization, which should be determined.
  3. In line 373-374, the Fig 3A is not clear, need to improve the quality of this picture.

Reviewer 3 Report

This study provided a novel oral way for anti-virus vaccinating in a commercial freshwater fish species. The intestinal bacteria Bacillus subtilis were used as a carrier of viral antigen and later stimulated the protection against GCRV-II. Though that the immune related mechanism was illustrated, the description about nutritional aspect were almost lacked. Also, both the intestinal colonization of mucosal spores and dose dependent effects were not accessed yet. In addition, both English writing and structural logics were not adequate. Thus, a major revision is required.

Major problems:

  1. There lacked an introduction about why using the immune indexes (complement C3, lysozyme and total superoxide dismutase) and why analyzed the expression of current used immune-related genes (IL-1β, TNF-α, IFN1 and MHC-II). Also, why analyze the IgM. I found some information in the discussion. But these should be told before the methods. The background for both systemic immune response and intestinal mucosal immune response upon GCRV-II infection should be included in the introduction, as a separate paragraph.
  2. Since that the feeds containing the genetically engineered Bacillus were the base for current study, the diet formula and how to prepare the Bacillus subtilis included feeds should be provided. Therefore, the introduction, methods, results and even discussion sections should include related contents accordingly.
  3. The dose-dependent effect of the genetically engineered Bacillus was lacked in this study. The dietary inclusion of the Bacillus should have parallel groups for the dose.
  4. Lines 241-242: What do you mean for “Oral grass carp”? The group name was very confusing.
  5. Lines 234-235: The formula and the protocol of making the diet should be detailed. Since that the particle surface’s area would be important for attaching of spores, the particle size of the feeds should be included.
  6. Lines 235-239: The condition for drying the feeds containing the spores was yet not described.
  7. Since that the successful oral immunization should be based on effectively consuming of feeds, the food intake should be measured for both the control and experimental diets. This would guarantee the effectively antigen up-taking for the VP56.
  8. Line 269: There should be descriptions or references for the used virus strain.
  9. Line 277: Please provide the exact living time for each dead fish, as supplementary data. This is important to compare the protective effects among different groups.
  10. Current qPCR data to confirm colonization of spores was not enough. The immunohistochemistry or fluorescence labeled method should be imported to directly evident the existence of intestinal mucosal spores.
  11. Line 440-441: Again, the group names were misleading.
  12. Figure 7: There should be a quantitative analysis for all the arrows pointed signs.

Minor problems:

  1. The English should be checked by a native speaker.
  2. Line 244: The present tense should be replaced with the past tense.
  3. Line 269: Did you mean 106 for “106”? Please check all superiors in the text.
  4. The “P” represented p value should be italic. Please check this throughout the text.
  5. For all the mathematical symbols, such as “<”, there should be a space either before or after the symbol. Please check this throughout the text.
  6. Lines 448-449: The italic format should be used for all gene names. Please check this point throughout the text.
  7. How ImageJ software was used to analysis the colonization efficiency was not provided in the methods. What was the version of current used ImageJ software? What formula and statistical method were used?

Round 2

Reviewer 3 Report

This revision has been improved at many aspects, especially the introduction about immune index and oral antigen’s immune effect, as well as the description of dossal effect in the result. However, some issues still remain.

Major problems:

  1. The protocol for making the diet has not been provided.
  2. The method to calculate the pathological features in both spleen and kidney could not be found in the text. The related method should be provided in section 2.14.

Minor problems:

  1. The full name of RPS should be provided upon first appearance rather than provided in the title of Table S4.
  2. The letter "p" was used randomly in uppercase, lowercase, normal, or italic! Please keep the format consistent throughout the text.
